# Adaptive Learning with Unknown Information Flows

**Yonatan Gur**
Graduate School of Business
Stanford University
Stanford, CA 94305
ygur@stanford.edu

**Ahmadreza Momeni**
Electrical Engineering Department
Stanford University
Stanford, CA 94305
amomenis@stanford.edu

## Abstract

An agent facing sequential decisions that are characterized by partial feedback needs to strike a balance between maximizing immediate payoffs based on available information, and acquiring new information that may be essential for maximizing future payoffs. This trade-off is captured by the multi-armed bandit (MAB) framework that has been studied and applied when at each time epoch payoff observations are collected on the actions that are selected at that epoch. In this paper we introduce a new, generalized MAB formulation in which additional information on each arm may appear arbitrarily throughout the decision horizon, and study the impact of such information flows on the achievable performance and the design of efficient decision-making policies. By obtaining matching lower and upper bounds, we characterize the (regret) complexity of this family of MAB problems as a function of the information flows. We introduce an adaptive exploration policy that, without any prior knowledge of the information arrival process, attains the best performance (in terms of regret rate) that is achievable when the information arrival process is a priori known. Our policy uses dynamically customized *virtual time* indexes to endogenously control the exploration rate based on the realized information arrival process.

## 1 Introduction

**Background and motivation.** In the presence of uncertainty and partial feedback on payoffs, an agent that faces a sequence of decisions needs to strike a balance between maximizing instantaneous performance and collecting valuable information that is essential for optimizing future decisions. A well-studied framework that captures this trade-off between new information acquisition (*exploration*), and optimizing payoffs based on available information (*exploitation*) is the one of multi-armed bandits (MAB) that first emerged in [20] in the context of drug testing, and was later extended by [15] to a more general setting. In this framework, an agent repeatedly chooses between $K$ arms where at each trial the agent pulls one arm and then receives a reward. In this formulation (known as the stochastic MAB setting), rewards are assumed to be identically distributed for each arm and independent across trails and arms. The objective of the agent is to maximize the cumulative return over a certain time horizon, and the performance criterion is the so-called *regret*: the expected difference between the cumulative reward received by the agent and the reward accumulated by a hypothetical benchmark, referred to as *oracle*, who holds prior information about the reward distribution of each arm (and thus repeatedly selects the arm with the highest expected reward). A sharp regret characterization for this traditional framework was first established by [13], followed by analysis of important policies such as $\epsilon$-greedy, UCB1, and Thompson sampling; see, e.g., [3], as well as [1]. The MAB framework focuses on balancing exploration and exploitation, typically under very little assumptions on the distribution of rewards, but with very specific assumptions on the future information collection process. In particular, optimal policy design is typically predicated on the assumption that at each period a reward observation is collected only on the arm that is selected by the policy at that time period (exceptions

to this common information structure will be discussed below). In that sense, such policy design does not account for information (e.g., that may arrive between pulls) that may be available in many practical settings, and that might be fundamental for achieving good performance.

In the current paper we relax the information structure of these classical frameworks by allowing arbitrary information arrival processes. Our focus is on: $(i)$ studying the impact of the information arrival characteristics (such as frequency and timing) on the achievable performance and on the manner in which a decision maker should balance exploration and exploitation; and $(ii)$ on adapting to a priori unknown sample path of information arrivals in real time. In that respect, we identify conditions on the information arrival process that guarantee the optimality of myopic policies (e.g., ones that at each period pull the arm with the highest estimated mean reward), and further identify adaptive MAB policies that guarantee the "best of all worlds" in the sense of establishing (near) optimal performance without prior knowledge of the information collection process.

**Main contributions.** On the modeling front, we introduce a new, generalized MAB formulation that relaxes strong assumptions classical MAB settings typically make on the information collection process. Our formulation considers information flows that correspond to the different arms and allows information to arrive at arbitrary rate and time, and therefore captures a large variety of real-world phenomena, yet maintains mathematical tractability.

On the analysis front, we establish lower bounds on the performance that is achievable by *any* non-anticipating policy in the presence of unknown information flows. We further show that our lower bounds can be achieved through suitable policy design. These results identify the minimax regret rate associated with the MAB problem with arbitrary information flows, as a function of the horizon length, the information arrival process, the number of arms, and parameters of the family of reward distributions. In particular, we obtain a *spectrum* of minimax regret rates that ranges from the classical regret rates that appear in the stochastic MAB literature when there is no or very little auxiliary information, to a constant regret (independent of the decision horizon length) when information arrives frequently and/or early enough.

We introduce an *adaptive exploration* policy that, without any prior knowledge of the auxiliary information flows, approximates the best performance that is achievable when the information arrival process is known in advance. This "best of all worlds" type of guarantee implies that rate-optimality is achieved uniformly over the general class of information flows at hand (including the case with no information flows, where classical guarantees are recovered). Our approach relies on using *endogenous* exploration rates that depend on the amount of information that becomes available over time. In particular, it is based on adjusting in real time the effective exploration rate through *virtual time* indexes that are dynamically updated based on information arrivals.

**Related work.** Several MAB settings have introduced cases where exploration is unnecessary (as a myopic decision-making policy may achieve optimal performance), versus cases where exploration should be done at an especially higher rate in order to maintain optimality. For example, [6] consider the contextual bandit framework and show that if the distribution of the contextual information guarantees sufficient diversity, then exploration becomes unnecessary and greedy policies can benefit from the natural exploration that is embedded in the information diversity to achieve asymptotic optimality. On the other hand, [7] considers a general MAB framework where the underlying reward distribution may change over time according to a budget of variation, and characterize the manner in which optimal exploration rates increase as a function of said budget. In addition, [18] consider a platform in which the preferences of arriving users may be biased by the experience of previous users and show that classical MAB policies may under-explore in this setting. These studies demonstrate that the extent of exploration that is required to maintain optimality strongly depends on particular problem characteristics that may often be a priori unknown to the decision maker. This introduces the challenge of endogenizing exploration to identify the appropriate rate of exploration and to approximate the best performance that is achievable under ex ante knowledge of the underlying problem characteristics. We address this challenge from information collection perspective. We identify conditions on the information arrival process that guarantee the optimality of myopic policies (e.g., ones that at each period pull the arm with the highest estimated mean reward), and further identify adaptive MAB policies that guarantee (near) optimal performance ("best of all worlds") without prior knowledge on the information arrival process.

In addition, few papers have considered regulating exploration based on a priori known characteristics in settings that are different than ours. For example, [21] consider regulating exploration and

exploitation in a setting where rewards are scaled by an exogenous multiplier that temporally evolves in an a priori known manner, and show that in such setting the performance of known MAB policies can be improved if exploitation is enhanced during periods with higher reward, and more exploration occurs in periods of low reward. Another approach of regulating exploration is studied by [12] in a setting that includes lock-up periods in which the agent cannot change her actions.

While in traditional MAB formulations in each time period observations are obtained only for the arm that was selected at that period, there are MAB formulations (and other sequential decision frameworks) in which more information can be observed in each time period. One important MAB framework where at each round the decision maker may collect some information on arms that were not pulled is the so-called contextual MAB setting, also referred to as bandit problem with side observations [23], or the associative bandit problem [19], where at each trial the decision maker observes a context carrying *information* about the arms. Another important example is the full-information *adversarial* MAB setting, where rewards are not characterized by a stationary stochastic process but are rather arbitrary and can be even selected by an adversary ([4], and [9]). In the full-information setting, at each time period, after pulling an arm, the agent observes the rewards generated by all the arms. While the adversarial nature of the latter makes it fundamentally different, in terms of achievable performance, analysis, and policy design, from the stochastic formulation that is considered here, it is also important to mention that the settings above still consider very specific information structures that are a priori known to the agent, as opposed to our setting where information flows are arbitrary and a priori unknown.

One of the challenges we address is to design a policy that adapts to unknown problem characteristics, in the sense of achieving ex-post performance that is as good (or nearly as good) as the one achievable under ex-ante knowledge of the information arrival process. This challenge dates back to studies in the statistics literature (see [22] and references therein), and has seen recent interest in the machine learning literature; examples include [17] that presents an algorithm that achieves (near) optimal performance in both stochastic and adversarial MAB regimes without prior knowledge of the nature of environment, [16] that considers an online convex optimization setting and derive algorithms that are rate optimal regardless of whether the target function is weakly or strongly convex, and [11] that studies the design of an optimal adaptive algorithm competing against dynamic benchmarks.

## 2  Formulation

Let $\mathcal{K} = \{1, \ldots, K\}$ be a set of arms and let $\mathcal{T} = \{1, \ldots, T\}$ denote a sequence of decision epochs. At each time period $t$, a decision maker selects one of the $K$ arms. When selecting arm $k \in \mathcal{K}$ at time $t \in \mathcal{T}$, a reward $X_{k,t} \in \mathbb{R}$ is realized and observed. For each $t \in \mathcal{T}$ and $k \in \mathcal{K}$, the reward $X_{k,t}$ is assumed to be independently drawn from some $\sigma^2$-sub-Gaussian distribution with mean $\mu_k$.[1] We denote the profile of rewards at time $t$ by $\mathbf{X}_t = (X_{1,t}, \ldots, X_{K,t})^\top$ and the profile of mean-rewards by $\boldsymbol{\mu} = (\mu_1, \ldots, \mu_K)^\top$. We further denote by $\boldsymbol{\nu} = (\nu_1, \ldots, \nu_K)^\top$ the distribution of the rewards profile $\mathbf{X}_t$. We assume that rewards are independent across the time periods and arms. We denote the highest expected reward and the best arm by $\mu^* = \max_{k \in \mathcal{K}} \mu_k$ and $k^* = \arg\max_{k \in \mathcal{K}} \mu_k$, respectively.[2] We denote by $\Delta_k = \mu^* - \mu_k$ the difference between the expected reward of the best arm, and the one associated with arm $k$. We assume that a priori known positive lower bound $0 < \Delta \leq \min_{k \in \mathcal{K} \setminus \{k^*\}} \Delta_k$ as well as a positive number $\sigma > 0$ for which all the reward distributions are $\sigma^2$-sub-Gaussian, and denote by $\mathcal{S} = \mathcal{S}(\Delta, \sigma^2)$ the class of $\Delta$-separated $\sigma^2$-sub-Gaussian distribution profiles:

$$\mathcal{S}(\Delta, \sigma^2) := \left\{ \boldsymbol{\nu} \;\middle|\; \Delta \cdot \mathbb{1}\{k \neq k^*\} \leq \Delta_k \text{ and } \mathbb{E}\left[e^{\lambda(X_{k,1} - \mu_k)}\right] \leq e^{\sigma^2 \lambda^2/2} \quad \forall k \in \mathcal{K}, \forall \lambda \in \mathbb{R} \right\}.$$

**Auxiliary information flows.** Before each round $t$, the agent may or may not observe reward realizations of some of the arms without pulling them. Let $\eta_{k,t} \in \{0, 1\}$ denote the indicator of observing an auxiliary information on arm $k$ at time $t$. We denote by $\boldsymbol{\eta}_t = (\eta_{1,t}, \ldots, \eta_{K,t})^\top$ the vector of indicators $\eta_{k,t}$'s at time step $t$, and by $\mathbf{H} = (\boldsymbol{\eta}_1, \ldots, \boldsymbol{\eta}_T)$ the information arrival matrix

with columns $\boldsymbol{\eta}_t$'s; we assume that this matrix is independent of the policy's actions and observations. If $\eta_{k,t} = 1$, then a random variable $Y_{k,t} \sim \nu_k$ is observed. We denote $\mathbf{Y}_t = (Y_{1,t}, \ldots, Y_{K,t})^\top$, and assume that the random variables $Y_{k,t}$ are independent across time periods and arms and are also independent from the reward realizations $X_{k,t}$. We denote the vector of information received at time $t$ by $\mathbf{Z}_t = (Z_{1,t}, \ldots, Z_{K,t})^\top$ where for any $k$ one has $Z_{k,t} = \eta_{k,t} \cdot Y_{k,t}$.

**Admissible policies, performance, and regret.** Let $U$ be a random variable defined over a probability space $(\mathbb{U}, \mathcal{U}, \mathbf{P}_u)$. Let $\pi_t : \mathbb{R}^{t-1} \times \mathbb{R}^{K \times t} \times \{0, 1\}^{K \times t} \times \mathbb{U} \to \mathcal{K}$ for $t = 1, 2, 3, \ldots$ be measurable functions; with some abuse of notation we aksi denote by $\pi_t \in \mathcal{K}$ the action at time $t$ given by $\pi_t = \pi_t(X_{\pi_{t-1}, t-1}, \ldots, X_{\pi_1, 1}, \mathbf{Z}_t, \ldots, \mathbf{Z}_1, \boldsymbol{\eta}_t, \ldots, \boldsymbol{\eta}_1, U)$ for $t = 1, 2, 3, \ldots$.

The mappings $\{\pi_t : t = 1, \ldots, T\}$, together with the distribution $\mathbf{P}_u$ define the class of admissible policies. We denote this class by $\mathcal{P}$. Note that policies in $\mathcal{P}$ depend only on the past history of actions and observations as well as auxiliary information arrivals, and allow for randomized strategies via their dependence on $U$. To evaluate the guaranteed performance of a policy $\pi \in \mathcal{P}$ under information arrival process $\mathbf{H}$ by the worst-case expected regret it incurs relative to the performance of an oracle that selects the arm with the highest expected reward, we define *regret* as follows:

$$\mathcal{R}_{\mathcal{S}}^\pi(\mathbf{H}, T) = \sup_{\boldsymbol{\nu} \in \mathcal{S}} \mathbb{E}_{\boldsymbol{\nu}}^\pi \left[ \sum_{t=1}^T (\mu^* - \mu^{\pi_t}) \right],$$

where the expectation $\mathbb{E}_{\boldsymbol{\nu}}^\pi[\cdot]$ is taken with respect to the noisy rewards, as well as to the policy's actions (throughout the paper we will denote by $\mathbb{P}_{\boldsymbol{\nu}}^\pi$, $\mathbb{E}_{\boldsymbol{\nu}}^\pi$, and $\mathcal{R}_{\boldsymbol{\nu}}^\pi$ the probability, expectation, and regret when the arms are selected according to policy $\pi$ and rewards are distributed according to $\boldsymbol{\nu}$).

**Discussion of model assumptions.** For the sake of simplicity, our model is based on a simple and well studied MAB framework [13]; However, we note that our methods and analysis can be directly applied to more general MAB frameworks such as the contextual MAB framework where mean rewards are linearly dependent on context vectors; see, e.g., [10] and references therein.

For the sake of simplicity of the model description, we assume that only one information arrival can occur before each time step for each arm (that is, for each time $t$ and arm $k$, one has that $\eta_{k,t} \in \{0, 1\}$). Notably, all our results hold for the case with more than one information arrival per time step per arm.

In our formulation we focus on auxiliary observations that have the same distribution as reward observations, but all our results hold for a broad family of information structures as long as unbiased estimators of mean rewards can be constructed from the auxiliary observations, that is, when there exists a mapping $\phi(\cdot)$ such that $\mathbb{E}[\phi(Y_{k,t})] = \mu_k$ for each $k$.

# 3 The impact of information flows on achievable performance

In this section we study the impact of auxiliary information flows on the performance that one could aspire to achieve. Our first result formalizes what *cannot* be achieved, establishing a lower bound on the best achievable performance as a function of the information arrival process.

**Theorem 1. (Lower bound on the best achievable performance)** *For any $T \geq 1$ and information arrival matrix $\mathbf{H}$, the worst case regret for any admissible policy $\pi \in \mathcal{P}$ is bounded below as follows*

$$\mathcal{R}_{\mathcal{S}}^\pi(\mathbf{H}, T) \geq \frac{C_1}{\Delta} \sum_{k=1}^K \log \left( \frac{C_2 \Delta^2}{K} \sum_{t=1}^T \exp \left( -C_3 \Delta^2 \sum_{s=1}^t \eta_{s,k} \right) \right),$$

*where $C_1$, $C_2$, and $C_3$ are positive constants that only depend on $\sigma$.*

Theorem 1 establishes a lower bound on the achievable performance in the presence of auxiliary information flows. The Theorem provides a *spectrum* of bounds on achievable performances, mapping many potential information arrival trajectories to the best performance they may allow. In particular, when $\mathbf{H} = 0$, we recover a lower bound of order $\frac{K}{\Delta} \log T$ that coincides with the lower bound established in [13], and [8] for the classical MAB setting. Theorem 1 further establishes that in the presence of auxiliary information flows regret rates may be lower relative to classical regret rates, and that the impact of information arrivals on the achievable performance depends on the *frequency* of these arrivals and on the *time* at which these arrivals occur; We discuss these observations in §3.1.

**Key ideas in the proof.** The proof of Theorem 1 adapts to our framework ideas of identifying a worst-case nature "strategy"; see, e.g. proof of Theorem 6 in [8]. While the full proof appears in the full version of the paper, we next illustrate its key ideas using the special case of two arms. Consider two possible profiles of reward distributions, $\boldsymbol{\nu}$ and $\boldsymbol{\nu}'$, that are "close" enough in the sense that it is hard to distinguish between the two, but "separated" enough such that a considerable regret may be incurred when the "correct" profile of distributions is misidentified. In particular, assume that the decision maker is a priori informed that the first arm generates rewards according to a normal distribution with standard variation $\sigma$ and a mean that is either $-\Delta$ (according to $\boldsymbol{\nu}$) or $+\Delta$ (according to $\boldsymbol{\nu}'$), and the second arm is known to generate rewards with normal distribution of standard variation $\sigma$ and mean zero for both $\boldsymbol{\nu}$, and $\boldsymbol{\nu}'$. To quantify a notion of distance between the possible profiles of reward distributions we use the Kullback-Leibler (KL) divergence. Using Lemma 2.6 from [22] that connects the KL divergence to error probabilities, we establish that at each period $t$ the probability of selecting a suboptimal arm must be at least $p_t^{\text{sub}} = \frac{1}{4} \exp\left(-\frac{2\Delta^2}{\sigma^2}\left(\mathbb{E}_{\boldsymbol{\nu}}[\tilde{n}_{1,T}] + \sum_{s=1}^{t} \eta_{1,s}\right)\right)$, where $\tilde{n}_{1,t}$ denotes the number of times the first arm is pulled up to time $t$ by the policy. Each selection of suboptimal arm contributes $\Delta$ to the regret, and therefore the cumulative regret must be at least $\Delta \sum_{t=1}^{T} p_t^{\text{sub}}$. We observe that if arm 1 has mean reward $-\Delta$, the cumulative regret must also be at least $\Delta \cdot \mathbb{E}_{\boldsymbol{\nu}}[\tilde{n}_{1,T}]$. Therefore the regret is lower bounded by $\frac{\Delta}{2}\left(\sum_{t=1}^{T} p_t^{\text{sub}} + \mathbb{E}_{\boldsymbol{\nu}}[\tilde{n}_{1,T}]\right)$ which is greater than $\frac{\sigma^2}{4\Delta} \log\left(\frac{\Delta^2}{2\sigma^2} \sum_{t=1}^{T} \exp\left(-\frac{2\Delta^2}{\sigma^2} \sum_{s=1}^{t} \eta_{1,s}\right)\right)$. The argument can be repeated by switching arms 1 and 2. For $K$ arms, we follow the above lines to establish $K$ lower bounds. Taking the average of these bounds, the result is established. $\qquad\square$

### 3.1 Discussion and subclasses of information flows

Theorem 1 demonstrates that auxiliary information flows may be leveraged to improve performance and reduce regret rates, and that their impact on the achievable performance increases when information arrivals are more frequent, and occur earlier. This observation is consistent with the following intuition: $(i)$ at early time periods we have collected only few observations and therefore the marginal impact of an additional observation on the stochastic error probabilities is relatively large; and $(ii)$ when information appears early on, there are more decision periods to come where this information can be used. To emphasize this observation we next demonstrate the implications on achievable performance of two concrete information arrival processes of natural interest: a process with a fixed arrival rate, and a process with a decreasing arrival rate.

**Stationary information flows.** Assume that $\eta_{k,t}$'s are i.i.d. Bernoulli random variables with mean $\lambda$. Then, for any $T \geq 1$ and admissible policy $\pi \in \mathcal{P}$, one obtains the following lower bound for the achievable performance. If $\lambda \leq \frac{\sigma^2}{4\Delta^2 T}$, then

$$\mathbb{E}_{\boldsymbol{H}}\left[\mathcal{R}_{\mathcal{S}}^{\pi}(\boldsymbol{H}, T)\right] \geq \frac{\sigma^2(K-1)}{4\Delta} \log\left(\frac{(1 - e^{-1/2})\Delta^2 T}{\sigma^2 K}\right),$$

and if $\lambda \geq \frac{\sigma^2}{4\Delta^2 T}$, then

$$\mathbb{E}_{\boldsymbol{H}}\left[\mathcal{R}_{\mathcal{S}}^{\pi}(\boldsymbol{H}, T)\right] \geq \frac{\sigma^2(K-1)}{4\Delta} \log\left(\frac{1 - e^{-1/2}}{2\lambda K}\right).$$

This class considers stationary information flows in which information arrives at a constant rate $\lambda$ throughout the horizon. Analyzing this arrival process reveals two different regimes. When the arrival rate of information is small enough, auxiliary observations become essentially ineffective, and one recovers the performance bounds that were established for the classical stochastic MAB problem. In particular, as long as there are no more than order $\Delta^{-2}$ information arrivals over $T$ time periods, this information does not impact achievable regret rates.[3] When $\Delta$ is fixed and independent of the horizon length $T$, the lower bound scales logarithmically with $T$. When $\Delta$ can scale with $T$, a bound of order $\sqrt{T}$ is recovered when $\Delta$ is of order $T^{-1/2}$. In both cases, there are known policies (such as UCB1) that guarantee rate-optimal performance; for more details see policies, analysis, and discussion in [3].

On the other hand, when there are more than order $\Delta^{-2}$ observations over $T$ periods, the lower bound on the regret becomes a function of the arrival rate $\lambda$. When the arrival rate is independent of the

horizon length $T$, the regret is bounded by a constant that is independent of $T$, and a myopic policy is optimal. For more details, see the full version of the paper.

**Diminishing information flows.** Fix some $\kappa > 0$, and assume that $\eta_{k,t}$'s are random variables such that for each arm $k \in \mathcal{K}$ and at each time step $t$, $\mathbb{E}\left[\sum_{s=1}^{t} \eta_{k,s}\right] = \left\lfloor \frac{\sigma^2 \kappa}{2\Delta^2} \log t \right\rfloor$. Then, for any $T \geq 1$ and admissible policy $\pi \in \mathcal{P}$, one obtains the following lower bound for the achievable performance. If $\kappa < 1$ then:

$$\mathbb{E}_{\boldsymbol{H}}\left[\mathcal{R}_{\mathcal{S}}^{\pi}(\boldsymbol{H}, T)\right] \geq \frac{\sigma^2(K-1)}{4\Delta} \log\left(\frac{\Delta^2/K\sigma^2}{1-\kappa}\left((T+1)^{1-\kappa} - 1\right)\right),$$

and if $\kappa > 1$ then:

$$\mathbb{E}_{\boldsymbol{H}}\left[\mathcal{R}_{\mathcal{S}}^{\pi}(\boldsymbol{H}, T)\right] \geq \frac{\sigma^2(K-1)}{4\Delta} \log\left(\frac{\Delta^2/K\sigma^2}{\kappa-1}\left(1 - \frac{1}{(T+1)^{\kappa-1}}\right)\right).$$

This class considers diminishing information flows under which the expected number of information arrivals up to time $t$ is of order $\log t$. The example illustrates the impact of the *timing* of information arrivals on the achievable performance, and suggests that a constant regret may be achievable even when the rate of information arrivals is decreasing. Whenever $\kappa < 1$, the lower bound on the regret is logarithmic in $T$, and there are well-studied MAB policies (e.g., UCB1, Auer et al. 3) that guarantee rate-optimal performance. When $\kappa > 1$, the lower bound on the regret is a constant, and one may observe that when $\kappa$ is large enough a myopic policy is asymptotically optimal. (In the limit $\kappa \to 1$ the lower bound is of order $\log \log T$.) For more details, see the full version of the paper.

**Discussion.** One may contrast the subclasses of information flows described above by selecting $\kappa = \frac{2\Delta^2 \lambda T}{\sigma^2 \log T}$. Then, in both settings the total number of information arrivals for each arm is $\lambda T$. However, while in the first example the information arrival rate is fixed over the horizon, in the second example this arrival rate is higher in the beginning of the horizon and gradually decreasing over time. By further selecting $\lambda = \frac{\sigma^2 \log T}{\Delta^2 T}$ one obtains $\kappa = 2$. The lower bound under stationary information flows is then logarithmic in $T$ (establishing the impossibility of constant regret in that setting), but the lower bound under the diminishing information flows is constant and independent of $T$ (in the next section we will observe that constant regret is indeed achievable in that setting). This observation echoes the intuition that earlier observations have larger impact on achievable performance, as at early periods there is only little information that is available and therefore the marginal impact of an additional observation on the performance is larger, and since earlier information can be used for more decision periods (as the remaining horizon is longer).[4]

The analysis above demonstrates that optimal policy design and the best achievable performance depend on the information arrival process: while policies such as UCB1 and $\epsilon$-greedy may be rate-optimal in some cases, a myopic policy can achieve rate-optimal performance in other cases. However, the identification of a rate-optimal policy relies on *prior knowledge* of the information flow. Therefore, an important question one may ask is: How can a decision maker *adapt* to an arbitrary and unknown information arrival process in the sense of achieving (near) optimal performance without any prior knowledge of the information flow? We address this question in §4.

$$\mathbb{E}\left[\sum_{s=1}^{t} \eta_{k,s}\right] = \lambda T \frac{t^{1-\gamma} - 1}{T^{1-\gamma} - 1}.$$

The expected number of total information arrivals for each arm, $\lambda T$, is determined by the parameter $\lambda$. The concentration of arrivals, however, is governed by the parameter $\gamma$. When $\gamma = 0$ the arrival rate is constant, which corresponds to the subclass of stationary information flows. As $\gamma$ increases, information arrivals concentrate in the beginning of the horizon, and $\gamma \to 1$ leads to $\mathbb{E}\left[\sum_{s=1}^{t} \eta_{k,s}\right] = \lambda T \frac{\log t}{\log T}$, which corresponds to the subclass of diminishing information flows. Then, one may apply similar analysis to observe that when $\lambda T$ is of order $T^{1-\gamma}$ or more, the lower bound is a constant independent of $T$.

# 4 A near-optimal adaptive policy

In this section we suggest a policy that adapts to a priori unknown information flow. Before laying down the policy, we first demonstrate that classical policy design may fail to achieve the lower bound in Theorem 1 in the presence of unknown information flows.

**The inefficiency of naive adaptations of MAB policies.** Consider a simple approach of adapting classical MAB policies to account for information that arrived so far in calculating the estimates of mean rewards while maintaining the structure of the policy otherwise. Such an approach can be implemented easily using some well-known MAB policies such as UCB1 or $\epsilon$-greedy. A first observation is that the performance bounds that are analyzed for these policies (e.g., in Auer et al. 3) do not improve (as a function of the horizon length $T$) in the presence of unknown information flows. Moreover, it is possible to show through lower bounds on the guaranteed performance that these policies indeed achieve sub-optimal performance. To demonstrate this, consider the subclass of stationary information flows described in §3.1, with an arrival rate $\lambda$ that is very large compared to $\frac{\sigma^2}{4\Delta^2 T}$. In that case, we have seen that the regret lower bound becomes constant whenever the arrival rate $\lambda$ is independent of $T$. However, the $\epsilon$-greedy policy, employs an exploration rate that is independent of the number of observations obtained for each arms and therefore effectively incurs regret of order $\log T$ due to performing unnecessary exploration.

**A simple rate-optimal policy.** We provide a simple and deterministic *adaptive exploration* policy that includes the key elements that are essential for appropriately adjusting the exploration rate and achieving good performance in the presence of auxiliary information flows. In what follows, we denote by $n_{k,t}$, and $\bar{X}_{k,n_{k,t}}$ the number of times a sample from arm $k$ has been observed and the empirical average reward of arm $k$ up to time $t$, respectively, that is,

$$n_{k,t} = \eta_{k,t} + \sum_{s=1}^{t-1} \left( \eta_{k,s} + \mathbb{1}\{\pi_s = k\} \right), \qquad \bar{X}_{k,n_{k,t}} = \frac{\eta_{k,t}Y_{k,t} + \sum_{s=1}^{t-1} \left( \eta_{k,s}Y_{k,s} + \mathbb{1}\{\pi_s = k\}X_{k,s} \right)}{n_{k,t}}.$$

Consider the following policy:

---

**Adaptive exploration policy.** Input: a tuning parameter $c > 0$.

1. Set initial virtual times $\tau_{k,0} = 0$ for all $k \in \mathcal{K}$, and an exploration set $\mathcal{W}_0 = \mathcal{K}$.

2. At each period $t = 1, 2, \ldots, T$:

   (a) Observe the vectors $\boldsymbol{\eta}_t$, and $\mathbf{Z}_t$.

   - Advance virtual times: $\tau_{k,t} = (\tau_{k,t-1} + 1) \cdot \exp\left( \frac{\eta_{k,t}\Delta^2}{c\sigma^2} \right)$     for all $k \in \mathcal{K}$

   - Update the exploration set: $\mathcal{W}_t = \left\{ k \in \mathcal{K} \mid n_{k,t} < \frac{c\sigma^2}{\Delta^2} \log \tau_{k,t} \right\}$

   (b) If $\mathcal{W}_t$ is not empty, select an arm from $\mathcal{W}_t$ with the fewest observations: (*exploration*)
   $$\pi_t = \arg\min_{k \in \mathcal{W}_t} n_{k,t}.$$

   Otherwise, Select an arm with the highest estimated reward: (*exploitation*)
   $$\pi_t = \arg\max_{k \in \mathcal{K}} \bar{X}_{k,n_{k,t}}.$$

   (c) Receive and observe a reward $X_{\pi_t,t}$

---

Clearly $\pi \in \mathcal{P}$. At each time step $t$, the adaptive exploration policy checks whether for each arm $k$ the number of observations that has been collected so far (through arm pulls and auxiliary information together) exceeds a *dynamic* threshold that depends logarithmically on the virtual time $\tau_{k,t}$, that is, whether arm $k$ satisfies the condition $n_{k,t} \geq \frac{c\sigma^2}{\Delta^2} \log \tau_{k,t}$. If yes, the arm with the highest reward estimator $\bar{X}_{k,n_{k,t}}$ is pulled (exploitation). Otherwise, the arm with the fewest observations is pulled (exploration). This approach guarantees that enough observations have been collected from each arm such that a suboptimal arm will be selected with a probability of order $t^{-c/8}$ or less.

The adaptive exploration policy endogenizes a common principle of balancing exploration and exploitation, by which the exploration rate should be set to guarantee that the overall loss due to exploration would equal the expected loss due to misidentification of the best arm; see e.g., [3] and

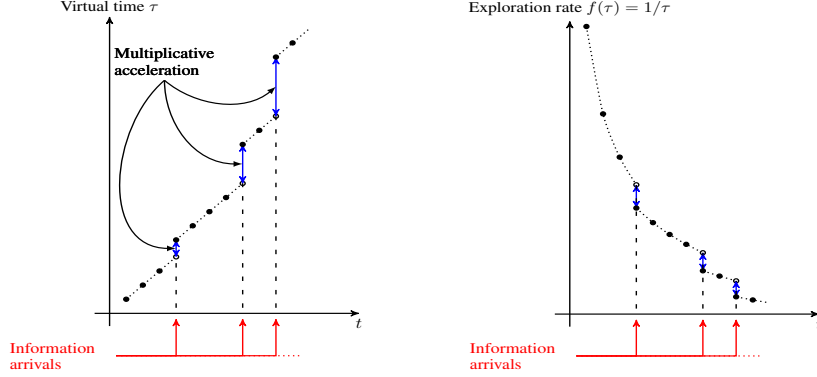

Figure 1: Illustration of the adaptive exploration policy. *(Left)* Virtual time index $\tau$ is advanced using multiplicative factors whenever auxiliary information is observed. *(Right)* Exploration rate decreases as a function of $\tau$, and in particular, exhibits discrete "jumps" whenever auxiliary information is observed.

references therein, the related concept of forced sampling in [14], as well as related discussions in [10] and [5]. In the absence of information flows, an exploration rate of order $1/t$ guarantees that the arm with the highest estimated mean reward can be suboptimal only with a probability of order $1/t$; see, e.g., the analysis of the $\epsilon$-greedy policy in [3], where at each time period $t$ exploration occurs uniformly at random with probability $1/t$. The manner in which the appropriate exploration rate decays captures the extent to which the value of new information deflates over time: at early time periods new information is more valuable as estimates are based on little information and there are many remaining decision epochs where this information could be used, and as time goes by new information becomes less valuable. In the presence of auxiliary information flows stochastic error rates decrease due to the additional observations. Our policy dynamically reacts to the information flows by reducing the exploration rate to guarantee that the loss due to exploration is balanced throughout the horizon with the expected loss due to misidentification of the best arm. The policy is doing so through adjusting the exploration rate of each arm based on a virtual time index $\tau_{k,t}$ associated with that arm, rather than based on the actual time period $t$ (which is appropriate in the absence of information flows). In particular, the adaptive exploration policy explores arm $k$ at a rate that would have been appropriate *without* auxiliary information flows at a *future* time step $\tau_{k,t}$. Every time additional information on arm $k$ is observed, a multiplicative factor is used to further advance the virtual time step $\tau_{k,t}$ by $\tau_{k,t} = (\tau_{k,t-1} + 1) \cdot \exp(\delta \cdot \eta_{k,t})$ for some suitably chosen $\delta$. The general idea of adapting the exploration rate of a policy by advancing a virtual time index as a function of the information arrival process is illustrated in Figure 1.

**Theorem 2. (Near optimality of the adaptive exploration policy)** *Let $\pi$ be the adaptive exploration policy with tuning parameter $c > 8$. For any $T \geq 1$ and auxiliary information arrival matrix $\mathbf{H}$:*

$$\mathcal{R}_{\mathcal{S}}^{\pi}(\mathbf{H}, T) \leq \sum_{k \in \mathcal{K}} \Delta_k \left( \frac{C_4}{\Delta^2} \log \left( \sum_{t=0}^{T} \exp \left( -\frac{\Delta^2}{C_4} \sum_{s=1}^{t} \eta_{k,s} \right) \right) + C_5 \right),$$

*where $C_4$ and $C_5$ are positive constants that depend only on $\sigma$.*

**Key ideas in the proof.** We decompose the overall regret into the regret over exploration time steps, and the regret over exploitation time steps. To bound the regret at exploration time periods we note that the virtual times could be expressed as $\tau_{k,t} = \sum_{s=1}^{t} \exp \left( \frac{\Delta^2}{c\sigma^2} \sum_{\tau=s}^{t} \eta_{k,\tau} \right)$, and that the expected number of observations from arm $k$ due to exploration and information flows together is at most $\frac{c\sigma^2}{\Delta^2} \log \tau_{k,T} + 1$. Subtracting the number of information arrivals $\sum_{t=1}^{T} \eta_{k,t}$ one obtains the first term in the upper bound. To bound the regret at exploitation time periods we use Chernoff-Hoeffding inequality to bound the probability that a sub-optimal arm has the highest estimated reward, given the minimal number of observations that must be collected on each arm. $\square$

The upper bound in Theorem 2 holds for any arbitrary sample path of information arrivals that is captured by the matrix $\mathbf{H}$, and matches the lower bound in Theorem 1 with respect to dependence on the time horizon $T$, as well as the sample path of information arrivals $\eta_{k,t}$'s, the number of arms $K$, and the minimum expected reward difference $\Delta$. In particular, this establishes a minimax regret rate of order $\frac{1}{\Delta} \sum_{k}^{K} \log \left( \sum_{t=0}^{T} \exp \left( -c \cdot \sum_{s=1}^{t} \eta_{k,s} \right) \right)$ for the MAB problem with auxiliary

information that is formulated here, where $c$ is a constant that may depend on problem parameters such as $K$, $\Delta$, and $\sigma$. Theorem 2 also implies that the adaptive exploration policy guarantees the best achievable regret (up to some multiplicative constant) under any arbitrary sample path of auxiliary information (and in particular, under the subclasses discussed in §3.1 for any values of $\lambda$ and $\kappa$).

## 5    Concluding remarks

In this study we considered a generalization of the stationary multi-armed bandits problem in the presence of unknown and arbitrary information flows on each arm. We studied the impact of such auxiliary information on the design of efficient learning policies and on the performance that can be achieved. In particular, we introduced an adaptive MAB policy that adapts in real time to the unknown information arrival process by controlling endogenizing the exploration rate through advancing virtual time indexes that are customized for each arm every time information on this arm arrives. We established that using this policy, one may guarantee the best performance (in terms of minimax regret) that is achievable under prior knowledge on the information arrival process.

## Footnotes

[1] A real-valued random variable $X$ is said to be sub-Gaussian if there is some $\sigma > 0$ such that for every $\lambda \in \mathbb{R}$ one has $\mathbb{E}e^{\lambda(X - \mathbb{E}X)} \leq e^{\sigma^2 \lambda^2/2}$.

[2] For the sake of simplicity, in the formulation and hereafter in the rest of the paper when using the $\arg\min$ and $\arg\max$ operators we assume that ties are broken in favor of the smaller index.

[3]This coincides with the observation that one requires order $\Delta^{-2}$ samples to distinguish between two distributions that are $\Delta$-separated; see, e.g., [2].

[4]This observation can be generalized by noting that the described subclasses are special cases of the following setting. Let $\eta_{k,t}$'s be independent random variables such that for each arm $k$ and every time period $t$, the expected number of information arrivals up to time $t$ satisfies

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
