[Reviews · NeurIPS 2018]

Reviewer 1



This is a good contribution. Posing the problem of under versus over exploration in terms of information flow is interesting. The authors clearly articulate the need to do this. The technical results are sensible and the presentation is good. There are a few comments that would help improve the presentation and the outreach of the paper (keeping in mind the variance in nips crowd). 1. Some real-world example need to be provided when motivating the stationary and diminishing information flows. 2. It is great that the authors summarized the technical results via remarks. However, some empirical plots would be helpful for different types/structures of H matrix? 3. There is no clear evaluation of the strategy to current adaptive MAB strategies -- understandable because of space, however, some comments about what to expect form such baseline comparison need to be reported.

Reviewer 2



Summary: The paper proposes a new multi-armed bandit (MAB) formulation, in which the agent may observe reward realizations of some of the arms arbitrarily before each round. The paper studies the impact of information flows on regret performance by deriving the regret lower bound with respect to information flows. Moreover, the paper proposes an adaptive exploration policy matching the regret lower bound. However, the insight under this setting is somewhat unclear to the reviewer. Therefore, the reviewer suggests voting for “weak accept”. Significance: The paper studies the problem of MAB with auxiliary information for some arms. The problem is interesting and has not been studied before. The paper provides an adaptive exploration policy with a near-optimal regret upper bound. However, in each round, it is assumed that the player obtains the knowledge of the arms with auxiliary information before choosing an arm to pull. It is hard to find the motivation for this assumption. Clarity: This paper is easy to follow. However, the clarity of the motivation can be further improved if the authors can give one or two real-world examples of this MAB formulation, especially for the case where the player observes the impact of information flows before choosing arms to pull. Another minor issue is that this paper does not have a conclusion section to summarize the contributions. Originality: The originality of this paper comes from the new, generalized MAB problem setting that considers auxiliary information of the arms may arrive arbitrarily between pulls. Technical Quality: The theoretical analyses are solid and the results are elegant. The algorithms are designed to keep the number of observations of a suboptimal arm be of order log(T), as the number of observations is unequal to the number of pulls. The proofs are regular and correct, with nothing new in the methodology.

Reviewer 3



The paper considers a new generalized formulation of multi-armed bandits (MAB) where additional information each arm may appear arbitrarily even when that arm is not pulled. The authors call this additional information as information flows. The paper is very well written. The authors derive a lower bound on regret and consider cases where the information flow is stationary (iid Bernoulli) and diminishing information flows. The authors propose a near-optimal policy that adapts to unknown information flow. The policy attempts to balance exploration and exploitation so that loss due to exploration would equal expected loss due to misidentification of the best arm, The paper is technically sound; the proofs seem correct and sketched well in the paper. My only regret is that there is no discussion on any practical implications of this model - for what sort of sequential decision making problems is this class models suitable?